# Efficacy of Different Doses of Daprodustat for Anemic Non-dialysis Patients with Chronic Kidney Disease: A Systematic Review and Network Meta-Analysis

**DOI:** 10.3390/jcm11102722

**Published:** 2022-05-11

**Authors:** Hammad Fadlalmola, Khaled Al-Sayaghi, Abdulqader Al-Hebshi, Maher Aljohani, Mohammed Albalawi, Ohoud Kashari, Alaa Alem, Mariam Alrasheedy, Saud Balelah, Faten Almuteri, Arwa Alyamani, Turki Alwasaidi

**Affiliations:** 1Department of Community Health Nursing, College of Nursing, Taibah University, Al-Madinah Al-Munawarah and KSA 1, Medina 42353, Saudi Arabia; 2Department of Medical Surgical Nursing, College of Nursing, Taibah University, Al-Madinah Al-Munawarah, KSA 1, Medina 42353, Saudi Arabia; kalsayaghi@yahoo.com; 3Nursing Division, Faculty of Medicine and Health Sciences, Sana’a University, Sana’a P.O. Box 1247, Yemen; 4Department of Pediatrics, Prince Mohammed Bin Abdulaziz Hospital, Ministry of National Guard Health Affairs, Medina 42353, Saudi Arabia; habshi05@hotmail.com; 5Department of Pathology, College of Medicine, Taibah University, Medina 42353, Saudi Arabia; maher.aljohani@yahoo.com; 6Department of Pathology and Laboratory Medicine, Prince Mohammed Bin Abdulaziz Hospital, Medina 42353, Saudi Arabia; 7Department of Medicine, College of Medicine, Taibah University, Medina 42353, Saudi Arabia; albalawi_21@hotmail.com (M.A.); dr.aa_alem@hotmail.com (A.A.); turkialwasaidi@hotmail.com (T.A.); 8Al Aziziyah Children Hospital, Jeddah 22253, Saudi Arabia; dr.o.kashari@gmail.com; 9Department of Pediatrics, East Jeddah Hospital, Jeddah 22253, Saudi Arabia; ma-rasheedy@moh.gov.sa; 10Department of Medicine, King Fahad Hospital, Medina 42353, Saudi Arabia; balila_ss@yahoo.com; 11Department of Pediatrics, King Salman Bin Abdulaziz Medical City, Madinah 42353, Saudi Arabia; dr-toonah@hotmail.com; 12Department of Oncology, King Saud Bin Abdulaziz Collage for Health Science, Jeddah 22253, Saudi Arabia; yamaniao@ngha.med.sa or; 13Princess Noorah Oncology Center, Ministry of National Guard Health Affairs, Jeddah 22253, Saudi Arabia; 14Prince Mohammed Bin Abdulaziz Hospital, Ministry of National Guard Health Affairs, Medina 42353, Saudi Arabia

**Keywords:** daprodustat, anemia, non-dialysis patients, kidney disease, systematic review

## Abstract

(1) Background: Anemia affects about 40% of patients with chronic kidney disease (CKD). Daprodustat improves serum hemoglobin in anemic patients by inhibiting prolyl hydroxylase of hypoxia-inducible factor. We conducted a network meta-analysis to investigate the direct and indirect effects of different doses of daprodustat compared to each other and erythropoietin and placebo. (2) Methods: We searched PubMed, Cochrane Library, Web of Science, and Scopus, for randomized clinical trials (RCTs) reporting data about different doses of daprodustat for anemia in nondialysis of CKDs. (3) Results: We eventually included five RCTs with a total sample size of 4566 patients. We found that the higher the dose of daprodustat, the greater the change in serum total iron binding capacity (TIBC), hemoglobin, and ferritin from baseline. Compared to placebo, daprodustat 25–30 mg was associated with the highest significant increase in serum hemoglobin (MD = 3.27, 95% CI = [1.89; 4.65]), a decrease in serum ferritin (MD = −241.77, 95% CI = [−365.45; −118.09]) and increase in serum TIBC (MD = 18.52, 95% CI = [12.17; 24.87]). (4) Conclusion: Higher daprodustat doses were associated with a higher impact on efficacy outcomes as serum total iron-binding capacity (TIBC), hemoglobin, and ferritin. However, data about the safety profile of different doses of daprodustat is still missing.

## 1. Introduction

Chronic kidney disease (CKD) was considered a major health burden that affected nearly 697.5 million people in 2017 [1]. Anemia was one of the most common complications of patients with CKD; 41% were reportedly anemic among 209,311 patients with CKD [2]. CKD anemia is usually caused by multiple factors, including a decreased red blood corpuscle (RBCs), an increased level of chronic inflammation-associated hepcidin that sequesters iron stores, and hypo-responsiveness induced by uremia to erythropoietin (EPO). However, reduced EPO level is considered the main factor that causes anemia of CKD [3,4,5,6]. Thus, anemia of CKD was treated mainly by recombinant human erythropoietin (rhEPO), EPO stimulating agents (ESAs), and iron supplements [7,8]. However, rhEPO and ESAs induced increment of serum hemoglobin over 13 g/dl were reported to be associated with severe adverse events, including stroke, venous thrombosis, cardiovascular compromises, and finally death [9,10,11,12]. Additionally, subcutaneous and parenteral administration of rhEPO, ESAs, and iron supplements was uncomfortable for patients.

Daprodustat is an orally administered active molecule that suppresses prolyl hydroxylase of hypoxia-inducible factor (HIF-PH). This stabilizes the HIF-α subunit and grants dimerization of HIF-α and HIF-β subunits [13,14] and simulates a state of hypoxia [15]. The state of hypoxia drives activation of transcription factors and genes, initiates the synthesis of EPO [16,17] and proteins that uptake, mobilize, and store iron, and modulates hepcidin [13,14,18,19].

Oral administration of daprodustat was reportedly associated with 12 times less serum EPO than the injection of rhEPO [20], yielding fewer cardiovascular morbidities and less need for iron supplementation [20,21].

A previous meta-analysis by Zheng et al. pooled data comparing the effect of daprodustat and control in anemia in patients and found that daprodustat could maintain hemoglobin with lower adverse events than rhEPO. However, they neither separated data on dialysis and non-dialysis patients nor accounted for different doses of daprodustat [22]. In addition, the previous meta-analysis compared the direct estimates of effect for only two arms at a time, in contrast to network meta-analyses that compare the efficacy of study arms directly and indirectly with no previous network meta-analysis (NMA) synthesized evidence about the safety and efficacy of all available daprodustat doses for anemia of CKDs in non-dialysis patients.

Therefore, we aimed to conduct an NMA to pool all analyzable data respecting the safety and efficacy of daprodustat, accounting for all different doses reported in the literature.

## 2. Materials and Methods

We completed this NMA following recommendations of the ‘Cochrane handbook’ [23] and reported it in accordance with the latest version of the preferred NMA reporting items of NMA [24,25].

### 2.1. Search Strategy and Data Collection

We executed a broad search in the subsequent databases: Scopus, PubMed, Cochrane Library, and Web of Science, using these search terms: ((Anemia OR erythrocytopenia) AND (non-dialysis) AND (Daprodustat OR Duvroq OR “GSK-1278863” OR “GSK1278863”)). Our search was updated in January 2022; the search term was modified respecting the recommendation of each database.

Retrieved records were gathered and searched for duplicates using EndNote. Two independent authors selected titles and full texts; disagreements were resolved by a third author.

### 2.2. Selection Criteria

We included studies that satisfied the subsequent criteria: (1) population: non-dialysis anemic patients with CKD; (2) intervention: daprodustat; (3) comparison: any conventional treatment for anemia of CKDs or placebo; (4) outcomes: serum hemoglobin (Hb), ferritin, total iron-binding capacity (TIBC), and serum iron; and (5) study designs were restricted to randomized clinical trials (RCTs) that reported data for each dose alone.

### 2.3. Data Extraction

We extracted data on:Characteristics of the enrolled population at the baseline and summary of the eligible trials including study ID (last name of first author/publication year), study arms, mean age, percentage of females, mean body mass index (BMI), hemoglobin, TIBC, ferritin, transferrin saturation (TSAT), hepcidin, race, baseline epoetin alfa dose, prior diseases, baseline eGFR, stage of CKDs, inclusion and exclusion criteria of eligible studies, time of follow-up, primary endpoints, and conclusions;Outcomes: serum HB, ferritin, TIBC, and serum iron;Domains of risk of the bias assessment tool.

### 2.4. Quality Assessment

We appraised the quality of the eligible studies by the Cochrane risk-of-bias tool (version one) that included the subsequent domains: sequence generated randomly, concealed allocation (blinded participants and personnel), blinded assessors of outcomes, incomplete data, selective reporting, and others [26]. Each domain was judged by two independent authors; any discrepancies were resolved by a third one.

### 2.5. Statistical Analysis

We used the netmeta package through RStudio to analyze this frequentist NMA. Continuous data were analyzed using mean difference (MD) and 95% confidence intervals (CI). We assessed substantial heterogeneity using the Chi-squared (Q2) test and quantified it using the I-squared test. Substantial heterogeneity was considered significant if the I2 test was valued more than 50% or the Q2 *p*-value was less than 0.1. Significant heterogeneity was treated by the random-effects model. The efficacy of the investigated study arms was sorted from highest to lowest in league tables.

## 3. Results

### 3.1. Literature Search Results

Our preliminary search retrieved 223 records; omission of duplicates resulted in 145 unique records that entered title and abstract screening. After the screening, five unique RCTs were eventually included [20,27,28,29,30]. The flow chart regarding the search strategy is shown in Figure 1.

### 3.2. Characteristics of the Enrolled Population at the Baseline and Summary of the Eligible Trials

This systematic review and NMA included five multicenter RCTs with a total sample size of 4566 patients that reported data about different doses of daprodustat and controls in anemic, non-dialysis dependent patients with CKDs. The included studies classified arms in which the mean age ranged from 54.8 to 71.3 years and the female percentage ranged from 36% to 84%. The range of follow-up duration widely from 4 to 52 weeks among the included RCTs. More information is exhibited in Appendix A.

We judge the bias risk in all included studies as low to moderate. All studies were reported with low bias in respect of selection, attrition, and reporting biases, with a higher risk in respect of other sources of bias domain. The summary of the risk of bias is presented in Figure 2.

### 3.3. Efficacy Outcomes

#### 3.3.1. Serum HB

We found that the change of HB from baseline was dose-dependent, the higher the dose, the higher the change. Compared to placebo, daprodustat 25–30 mg was associated with the highest significant change in serum HB (MD = 3.27, 95% CI = [1.89; 4.65]), followed by daprodustat 8–12 mg (MD = 2.48, 95% CI = [1.50; 3.46]) and daprodustat 5–6 mg (MD = 1.18, 95% CI = [0.51; 1.85]).

Furthermore, daprodustat 25–30 mg was associated with a significant higher change in serum HB than daprodustat 5–6 mg (MD = 2.09, 95% CI = [0.56; 3.62]), rhEPO (MD = 2.79, 95% CI = [1.23; 4.34]) and daprodustat 2–4 mg (MD = 2.92, 95% CI = [1.40; 4.44]).

Furthermore, daprodustat 8–12 mg was associated with a significant higher change in serum HB than daprodustat 5–6 mg (MD = 1.30, 95% CI = [0.12; 2.48]) rhEPO (MD = 2.00, 95% CI = [0.78; 3.21]) and daprodustat 2–4 mg (MD = 2.13, 95% CI = [0.96; 3.30]); and daprodustat 5–6 mg was significantly associated with a higher increase in serum HB than daprodustat 2–4 mg (MD = 0.83, 95% CI = [0.16; 1.50]) (Figure 3).

#### 3.3.2. Serum Ferritin

We found that the higher the daprodustat dose, the lower the serum ferritin from base line. Compared to placebo, daprodustat 25–30 mg was associated with the highest significant change in serum ferritin (MD = −241.77, 95% CI = [−365.45; −118.09]), followed by daprodustat 8–12 mg (MD = −92.30, 95% CI = [−154.42; −30.18]) and daprodustat 5–6 mg (MD = −77.50, 95% CI = [−125.01; −29.99]). Furthermore, daprodustat 25–30 mg was associated with a more significant change in serum ferritin than daprodustat 8–12 mg (MD = −149.47, 95% CI = [−265.71; −33.23]), daprodustat 5–6 mg (MD = −164.27, 95% CI = [−296.76; −31.78]), daprodustat 2–4 mg (MD = −257.87, 95% CI = [−436.77; −78.97]), and rhEPO (MD = -270.25, 95% CI = [−449.94; −90.56]) (Figure 4).

#### 3.3.3. Total Iron-Binding Capacity (TIBC)

We found that the higher the dose of daprodustat, the higher the change of TIBC from the baseline.

Compared to placebo, daprodustat 25–30 mg was associated with the highest significant change in TIBC (MD = 18.52, 95% CI = [12.17; 24.87]), followed by daprodustat 8–12 mg (MD = 16.20, 95% CI = [9.85; 22.55]), daprodustat 5–6 mg (MD = 8.80, 95% CI = [3.18; 14.42]), and daprodustat 2–4 mg (MD = 5.60, 95% CI = [0.26; 10.94]).

In addition, daprodustat 25–30 mg was significantly associated with higher change in TIBC than daprodustat 5–6 mg (MD = 9.72, 95% CI = [1.24; 18.20]), daprodustat 2–4 mg (MD = 12.92, 95% CI = [4.62; 21.22]), and rhEPO (MD = 17.79, 95% CI = [8.73; 26.86]); daprodustat 8–12 mg was associated with a significant higher change in TIBC than daprodustat 2–4 mg (MD = 10.60, 95% CI = [2.30; 18.90]) and rhEPO (MD = 15.47, 95% CI = [6.41; 24.53]); daprodustat 5–6 mg and daprodustat 2–4 mg was associated with a significant higher change in TIBC than rhEPO ((MD = 8.07, 95% CI = [1.35; 14.79]), (MD = 4.87, 95% CI = [1.22; 8.52]), respectively) (Figure 5).

#### 3.3.4. Serum Iron

We found that rhEPO was numerically but not significantly associated with a higher change in serum iron from the base line than daprodustat 5–6 mg (MD = −1.05, 95% CI = [−4.57; 2.46]), daprodustat 2–4 mg (MD = −1.05, 95% CI = [−2.26; 0.15]) and placebo (MD = −2.05, 95% CI = [−4.49; 0.38]) (Figure 6).

**Figure 4 jcm-11-02722-f004:**
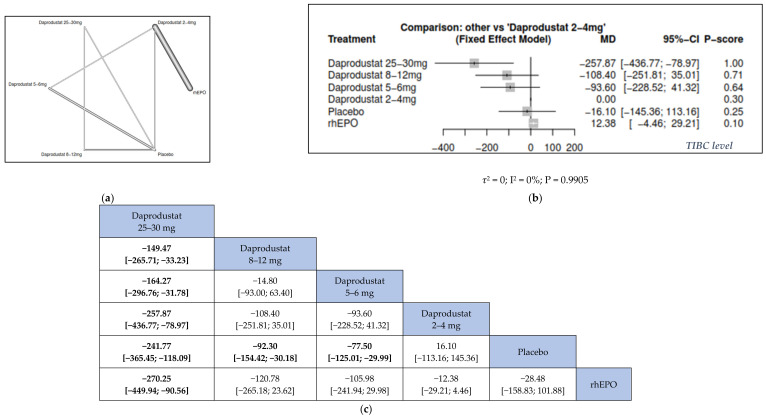
Network meta-analysis results of rate of serum ferritin. (**a**) Network graph showing direct evidence between the assessed interventions. (**b**) A forest plot was generated by comparing all interventions with “placebo”; P-score was used for ranking. (**c**) The league table represents the network meta-analysis estimates for all interventions comparisons; the results are the mean difference (MD) with 95% CI, bold items are statistically significant.

**Figure 5 jcm-11-02722-f005:**
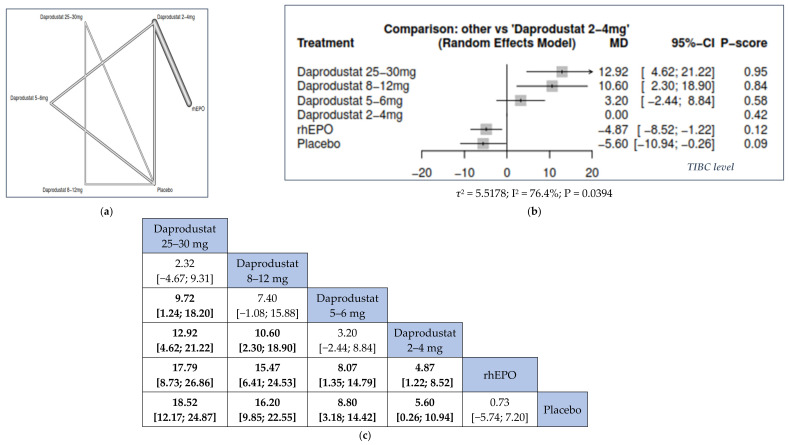
Network meta-analysis results of total iron-binding capacity (TIBC). (**a**) Network graph showing direct evidence between the assessed interventions. (**b**) A forest plot was generated by comparing all interventions with “placebo”; P-score was used for ranking. (**c**) The league table represents the network meta-analysis estimates for all interventions comparisons; the results are the mean difference (MD) with 95% CI, bold items are statistically significant.

**Figure 6 jcm-11-02722-f006:**
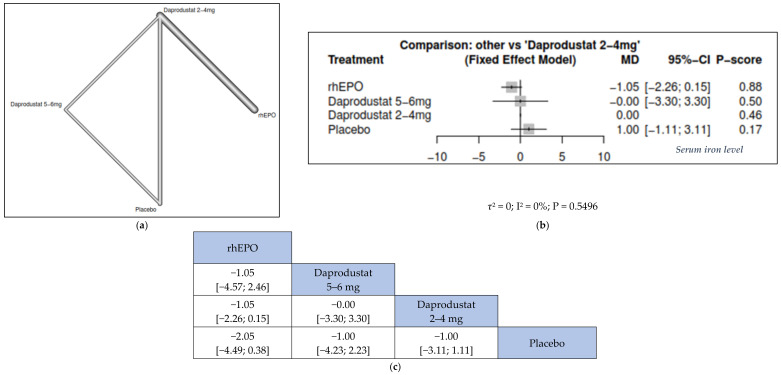
Network meta-analysis results of serum iron. (**a**) Network graph showing direct evidence between the assessed interventions. (**b**) A forest plot was generated by comparing all interventions with “placebo”; P-score was used for ranking. (**c**) The league table represents the network meta-analysis estimates for all interventions comparisons; the results are the mean difference (MD) with 95% CI, bold items are statistically significant.

#### 3.3.5. Lipid Level

Three studies have examined the effect of daprodustat on lipid levels [20,28,30]. Nangaku et al. reported minor changes in lipid parameters after daprodustat with 61% hyperlipidemic patients in the daprodustat group compared with 70% in the continuous erythropoietin receptor activator (CERA) group. In the daprodustat group, a slight decrease in total cholesterol, LDL cholesterol, and HDL cholesterol levels was detected. At the same time, HDL increased with little change in total, and LDL cholesterol was observed in the CERA group. Holdstock et al. 2016 reported slightly similar results with 76% hyperlipidemic patients in the daprodustat group compared with 78% in the placebo group. While in 2019, Holdstock et al. stated that 3% were hyperlipidemic in both groups; daprodustat and control.

#### 3.3.6. Cardiovascular (CV) Outcomes

Two included studies reported the effect of daprodustat on CV outcomes, especially major adverse cardiovascular events (MACE), such as all-cause mortality, non-fatal stroke, or non-fatal myocardial infarction [28,29]. Singh 2021 reported that 378 of 1937 patients (19.5%) in the daprodustat group and 371 of 1935 patients (19.2%) in the darbepoetin alfa group had MACE with a hazard ratio of 1.03; 95% CI, 0.89 to 1.19 without increased risk. Moreover, Holdstock 2019 stated that five (3%) of 170 patients in the daprodustat group had MACE compared with only one (1%) of 80 patients in the control group without sufficient data for comparison. 

### 3.4. Safety Outcomes

Brigandi et al. reported adverse events (AE) among 57% of non-dialysis dependent patients with stages 3-5 of CKD. Nausea, the most common adverse event, was reported in 21% and 40% of patients administered 25 mg and 100 mg of daprodustat, respectively. The higher doses of 50 and 100 mg produced an excessive erythropoietin response. So, the Hb response was also high, leading to AEs or drug discontinuation [27]. Similarly, the Holdstock 2016 study found that nausea was the most common adverse event associated with daprodustat, reported with two patients administered 0.5 mg and one patient-administered 5 mg doses [20].

Another study by Holdstock et al. found that the most common SAEs associated with daprodustat were nasopharyngitis, diarrhea, and nausea, without accounting for different doses of daprodustat [28]. A study by Singh et al. found that more than 43% of patients with CKD reported SAEs after initiation of daprodustat. However, SAEs were comparable between the daprodustat and control groups except for cancer and esophageal erosions, significantly higher in the daprodustat than in control groups [29]. Nangaku 2021 reported that after the 52-week treatment period, 92% of patients experienced ≥1 AE in the daprodustat group and 89% in the CERA group. The most common AE was nasopharyngitis and constipation. Serious AEs have appeared in 23% of patients with daprodustat and 29% with CERA. No significant difference between the treatment groups in AEs of special interest [30].

## 4. Discussion

Overall, daprodustat was associated with a significant increment of serum HB and TIBC, and a significant reduction of serum ferritin compared to placebo and rhEPO; the higher the dose of daprodustat, the higher the change of serum HB, TIBC, and ferritin from the baseline. However, we found no significant difference between the doses of daprodustat, rhEPO, and placebo regarding the change of serum iron from the baseline.

Although we pooled data about non-dialysis dependent patients, our results were consistent with what was reported by a previous meta-analysis by Zheng et al. that pooled data about both dialysis and non-dialysis dependent patients [22].

Daprodustat is a small active molecule administered orally, once daily, and in a dose dependent on the severity of anemia (baseline HB). Daprodustat inhibits hypoxia-inducible factor (HIF-PH) prolyl hydroxylase and mimics a hypoxic state. Inhibition of HIF-PH stabilizes HIF-α that dimerizes with HIF-β, eventually entering the nucleus to activate the EPO gene and stimulate erythropoiesis [31,32]. Among the isoforms of HIF-α, HIF-1α and HIF-2α are the most directly related to iron metabolism [33]; HIF-2α is the primary regulator of genes for iron metabolism in the liver [34]. HIF-2α was reported to be associated with the up-regulation of essential genes for iron absorption in the intestine that incorporates duodenal cytochrome-b and divalent transporters [34]. HIF-1α was reportedly associated with upregulation transferrin, which drives iron to the tissue [18,34]. Treatment with daprodustat is associated with the accumulation of both HIF1α and HIF2α subunits. 

Furthermore, HIF-1 was reported to upregulate vascular endothelial growth factor (VEGF) that stimulates angiogenesis and was associated with proliferative retinopathy and enhanced tumor growth [35,36]. However, Tsubakihara et al. followed patients who were administered daprodustat for 24 weeks and found no significant changes during fundus examination [37]; five studies supported these results [20,27,28,38,39]. 

Additionally, daprodustat mimicked hypoxia keeps the level of EPO in the physiological range, up to 17 times less than the level induced by rhEPO and ESA [20,40]. This might decrease cardiovascular compromises associated with supraphysiologic levels of EPO [41,42]. Brigandi et al. reported that the increment in serum EPO was proportional to the administered dose of daprodustat. But the high withdrawal rate associated with a high daprodustat dose resulted in a paradoxical lower response in the daprodustat group despite higher EPO levels [27]. Holdstock et al. also reported the dose-dependent relationship between daprodustat and serum EPO [20].

To our knowledge, we conducted the first network meta-analysis investigating direct and indirect comparisons of daprodustat, EPO, and placebo regarding the efficacy outcomes. Additionally, we pooled data exclusive for non-dialysis dependent patients and accounted for different doses of daprodustat reported in the literature. Our study included the most recent RCTs found in the literature with a combined largest sample size (4566 patients), approximately triple the number enrolled in the latest meta-analysis (1514 patients) [22].

Our study has some limitations. We did not pool data regarding safety outcomes and hepcidin levels due to missing data taking into account the different doses of daprodustat. The eligible studies generally enrolled a small sample size; more than 80% of the total study size was enrolled by one of the included studies [29]. Furthermore, the duration of follow-up in three of the included studies ranged from 4 to 24 weeks [20,27,28], which was not sufficient to observe long-term efficacy outcomes. All included studies were funded by GlaxoSmithKline (GSK) with a highly possible conflict of interest, yielding them all to be high in the other bias domain. Furthermore, the effect of daprodustat on oral/IV iron use was not reported by most of the included studies and could not be analyzed. Finally, our results are based only on a small sample size reported only by one study with short follow-up duration.

## 5. Conclusions

We found that daprodustat had a significant effect on elevating serum HB and TIBC and decreasing serum ferritin. Increased daprodustat dose (up to 25–30 mg) may be associated with increased effect. Due to lacking studies considering doses of daprodustat while reporting safety outcomes, we could not investigate the association between higher doses of daprodustat and safety parameters.

We recommend conducting high-quality multi-center RCTs by enrolling a large population number and considering different doses of daprodustat, especially larger doses. This will help synthesize more generalizable and trustable evidence about dose-dependent safety and efficacy parameters.

## Figures and Tables

**Figure 1 jcm-11-02722-f001:**
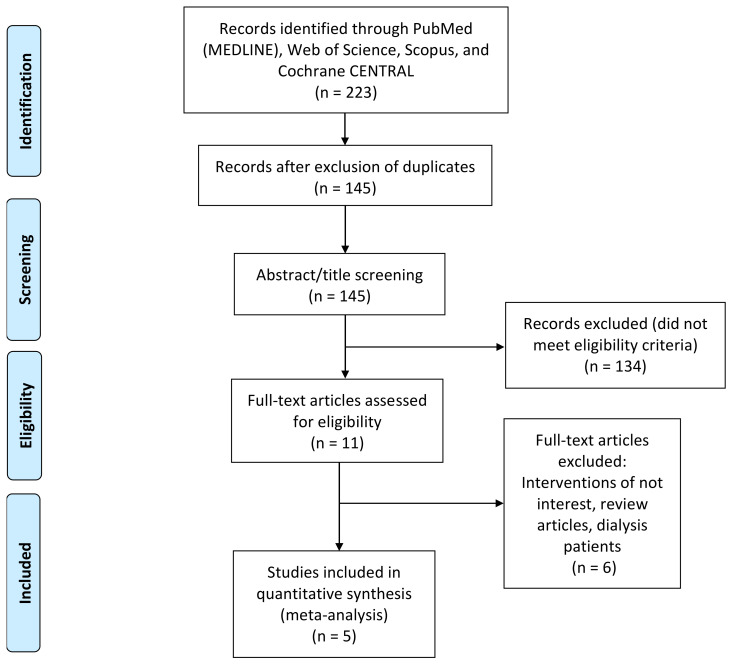
PRISMA flow diagram, which summarizes the literature search and the number of the obtained records.

**Figure 2 jcm-11-02722-f002:**
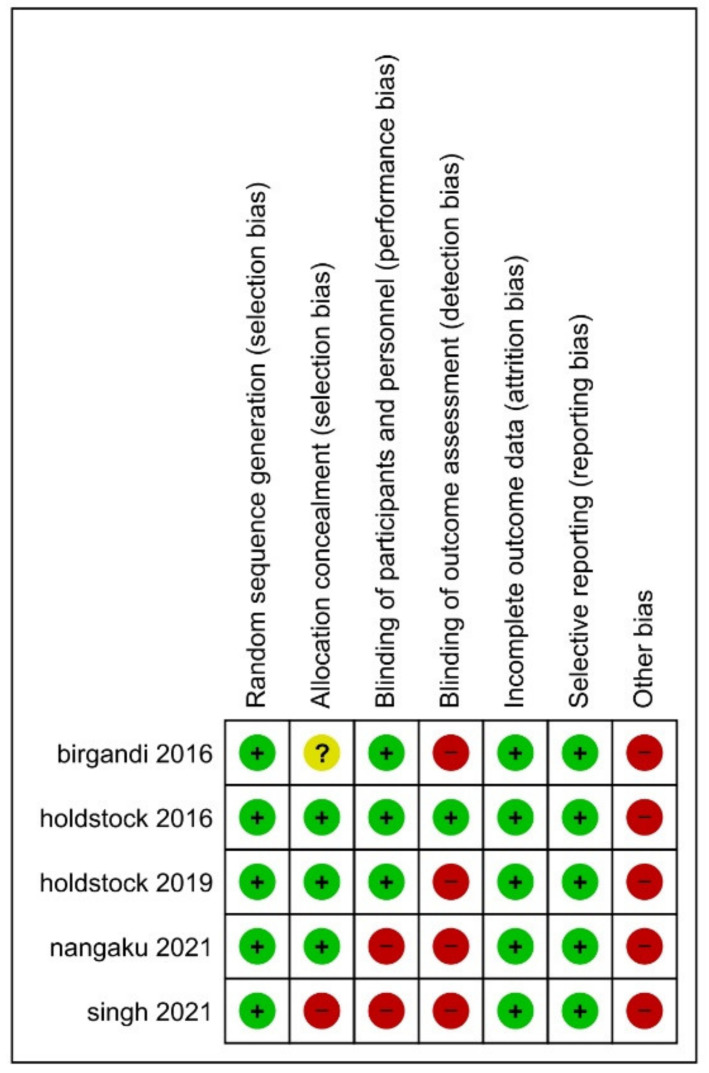
Risk of bias summary of the included studies.

**Figure 3 jcm-11-02722-f003:**
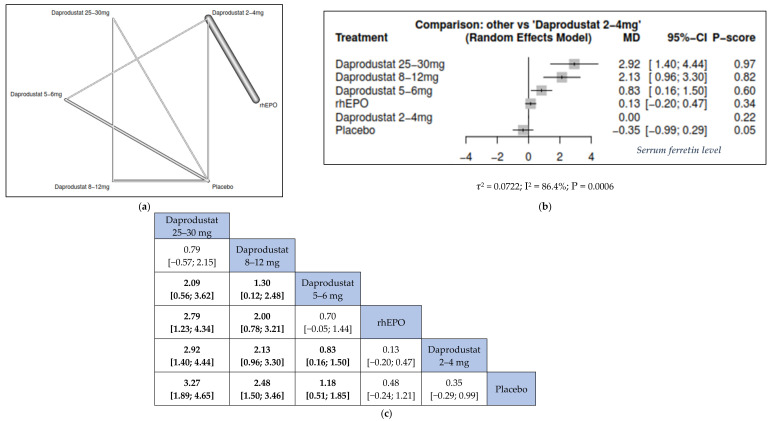
Network meta-analysis results of serum HB. (**a**) Network graph showing direct evidence between the assessed interventions. (**b**) A forest plot was generated by comparing all interventions with “placebo”; P-score was used for ranking. (**c**) The league table represents the network meta-analysis estimates for all interventions comparisons; the results are the mean difference (MD) with 95% CI, bold items are statistically significant.

## Data Availability

The datasets used and/or analyzed during the current study are available from the corresponding author on reasonable request.

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
