# Peer review of "Efficacy of Different Doses of Daprodustat for Anemic Non-dialysis Patients with Chronic Kidney Disease: A Systematic Review and Network Meta-Analysis"

_jcm, 2022, doi:10.3390/jcm11102722_

Round 1

Reviewer 1 Report

The authors conducted here a network meta-analysis that compares the efficacy of Daprodustat, a novel oral HIF-PH inhibitor, in non-dialysis CKD population.

Although the HIF-PHI are of great interest, the present study doesn’t bring novelty and have major flaws as no data on safety (specially CV outcomes) are analyzed.

Major concerns

1 / Regarding the efficacy of Daprodustat:

*The authors should analyze the effect of Daprodustat on oral/IV Iron use. Is there a benefit in terms on iron supplementation? This is of particular interest.

*As others HIF-PHI modulate Lipid level, it could be interesting to see whether a similar effect is observed with Daprodustat.

2/ Regarding the safety of Daprodustat

The section on the safety deserved to be more detailed. If the authors are not able to perform statistical analysis, they should at least talk about the CV profile/ safety. A table is required

3/ Regarding the discussion that is too short:

* The authors should discuss the rhythm and dose administration of Dapro as compared to ESAs

* The authors should discuss the specific inhibition of dapro on HIF sub unit (isoforms 1,2,3?)

* The paragraph on VEGF is not relevant as no data on vascular events are shown ( e/g CV events, or progression of specific disease  such as retinopathy). If the authors do not show this kind of result, I suggest to deleted this section.

Author Response

Dear Editor,

We are happy to receive the reviewers' comments which are excellent additions to the quality of our manuscript. We have made all the changes recommended by the reviewers. Below, we attach a table with all changes and our point-by-point response to reviews' comments.

Reviewer 2 Report

This is a systemic review & network meta-analysis about the efficacy of different doses of daprodustat for anemic non-dialysis patients with CKD. Based on five RCTs with a total sample size of 4566 patients, the authors found that the higher the dose of daprodustat, the greater the change in serum hemoglobin, TIBC, and ferritin and compared to placebo, daprodustat 25-30 mg was associated with the highest significant increases in serum hemoglobin. Therefore, they conclude that higher daprodustat doses were associated with a higher impact on efficacy outcome as serum TIBC and hemoglobin.

I think that this meta-analysis is well-designed, but there is the important problem in study population!!

On five RCTs, the maximal dose of daprodustat was 5mg in four RCTs with a sample size of 4496. Only one RCT (Brigandi 2016) with a sample size of 70 patients reported the efficacy of higher dose (more than 10mg) of daprodustat (10mg, 17 patients/25mg, 14 patients/50mg, 15 patients/100mg, 15 patients). Furthermore, in the discussion part of this RCT, there is a sentence like this “1) response rate was highest for the lower doses (10 and 25mg) in both groups unlike 50- and 100mg doses. 2) Interpretation of the response at higher doses (50 and 100 mg) is problematic because very few patients receiving these doses completes the study due to AEs leading to discontinuation.” In addition, although the minimal duration of F/U in other three RCTs with sample size of 4406 patients is 24 weeks, this RCT was only 8 weeks. Therefore, there is the limitation of evaluation for safety of daprodustat, especially higher doses.

In conclusion, this meta-analysis may have the risk of overestimation for the efficacy and safety of daprodustat due to the inclusion of very exceptional data.

# Minor revision: The name of X axis in Figure 4,5,6 (b) should be changed according to the contents of results.

(hemoglobin level -> Serum ferritin, TIBC, serum iron)

Author Response

(The authors gave the same response as above.)

Round 2

Reviewer 1 Report

The Authors partially responded to my concerns. 

1/ They still have to describe to CV outcomes that comes mostly from the study by singh (NEJM 2021). 

2/ regarding the lipid level sectionn : please, dont use the word hyperlipidemic but give the level values for each studies. 

Reviewer 2 Report

According to comments, this paper is well revised. I recommend for "accept in present form".